# Using Reinforcement Learning for Multi-Objective Cluster-Level NPI Optimization

Xueqiao Peng
peng.969@osu.edu
The Ohio State University
Columbus, Ohio, USA

Jiaqi Xu
xu.4015@osu.edu
The Ohio State University
Columbus, Ohio, USA

Xi Chen
chen.10183@osu.edu
The Ohio State University
Columbus, Ohio, USA

Dinh Song An Nguyen
nguyen.2687@osu.edu
The Ohio State University
Columbus, Ohio, USA

Andrew Perrault
perrault.17@osu.edu
The Ohio State University
Columbus, Ohio, USA

## ABSTRACT

Non-pharmaceutical interventions (NPIs) play a critical role in the defense against emerging pathogens. Among these interventions, familiar measures such as travel bans, event cancellations, social distancing, curfews, and lockdowns have become integral components of our response strategy. Contact tracing is especially widely adopted. However, the optimization of contact tracing involves navigating various trade-offs, including the simultaneous goals of minimizing virus transmission and reducing costs. Reinforcement learning (RL) techniques provides a promising avenue to model intricate decision-making processes and optimize policies to achieve specific objectives, but even modern deep RL techniques struggle in the high dimensional partially observable problem setting presented by contact tracing. We propose a novel RL approach to optimize a multi-objective infectious disease control policy that combines supervised learning with RL, allowing us to capitalize on the strengths of both techniques. Through extensive experimentation and evaluation, we show that our optimized policy surpasses the performance of five benchmark policies.

## KEYWORDS

reinforcement learning, machine learning, contact tracing, public health

**ACM Reference Format:**

Xueqiao Peng, Jiaqi Xu, Xi Chen, Dinh Song An Nguyen, and Andrew Perrault. 2023. Using Reinforcement Learning for Multi-Objective Cluster-Level NPI Optimization. In *epiDAMIK 2023: 6th epiDAMIK ACM SIGKDD International Workshop on Epidemiology meets Data Mining and Knowledge Discovery, August 7, 2023, Long Beach, CA, USA.* , 7 pages.

## 1 INTRODUCTION

The COVID-19 pandemic has highlighted the crucial role of non-pharmaceutical interventions (NPIs) in effectively managing the spread of infectious diseases. The implementation of NPIs requires careful consideration of multiple objectives, including the prevention of viral transmission and the reduction of costs associated with quarantine measures. Contact tracing has emerged as a widely adopted policy within the realm of NPIs and has been extensively studied in the context of COVID-19 [7, 8, 11, 21].

Nevertheless, optimizing NPIs remains a challenging open problem in many settings for several reasons. First, the objective is inherently multi-objective—intensified control efforts lead to higher costs. In addition, sensing actions, such as testing, may be included in all but the earliest stages of an infectious disease crisis. These have their own costs and constraints associated with them. Secondly, inferring the probability that an individual is difficult for infections that do substantial transmission asymptomatically, such as SARS-CoV-2. This inference problem is perhaps surprisingly high dimensional, as we show it is dependent on the symptom status and test results of all individuals in the same cluster due to the transmission heterogeneity.

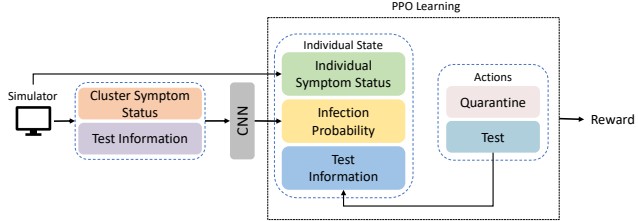

**Figure 1: Illustration of our approach. We combine a infection probability decoder that uses supervised learning with a reinforcement learning-based policy.**

In this work, our goal is to develop a generic approach for cluster-level optimization of NPIs. To tackle this challenge, we propose a novel approach that integrates convolutional neural networks (CNN) and reinforcement learning (RL) model[5, 20] (Fig. 1). The CNN is used to solve the high dimensional infection inference problem and uses a novel representation of the symptom and test state of the entire cluster as input, allowing a single CNN to be trained for all cluster sizes. The RL agent takes the CNN output and other features as its state and selects an action for each individual (including quarantine and testing) and aims to maximize a multi-objective reward function. This reward function includes a penalty

for days where an individual is infectious but not isolated, a penalty for days where they are quarantined but not infectious, as well as a cost for any control action that is taken (e.g., test cost). As a case study, we have developed a branching process-based SARS-CoV-2 virus simulator, where we evaluate the effectiveness of our method. In this work, we focus on optimization only—in the longer term, we aim to use the results of optimization to automatically discover simple, implementable policies.

This paper makes the following contributions:

- We propose a novel RL approach for finding optimal contact tracing policies. Our approach combines a supervised learning model with an RL model, leveraging the strengths of both techniques to optimize the desired objectives. The resulting agent can be trained and deployed simultaneously across all cluster sizes.
- We show the existence of a theoretically simple, yet optimal, *threshold* type policy for contact tracing in the setting where no sensing actions are available. Running this policy requires supervised learning only.
- We develop a simple branching process-based model for SARS-CoV-2 and compare our policies with baselines. We show that we achieve better rewards across a range of objective parameters.

*Related work.* We identify two main thrusts of work that optimize contact tracing and NPIs: network and branching process. Network models represent connections between individuals as edges in a possibly dynamic contact graph [4, 9, 12, 15, 16]. These approaches can leverage network structure in their decisions but make the strong assumption that the entire contact network is known. The closest existing approach to ours is RLGN [12], which formulates the problem as a sequential decision-making task within a temporal graph process. These approaches often consider a fixed budget of interventions rather than a multi-objective reward function. In contrast, branching processes are used, resulting in a cluster-based, tree-structured view of contagion [10, 13, 17]. These approaches have the advantage of aligning more closely with the information available to public health decision-makers in many practical settings (but allow for less expressive policies). All of these models are agent-based in the sense that they model individuals rather than subpopulations—because contact tracing decisions depend on the specific time that certain events happen for individuals (e.g., exposure, symptoms), the additional detail that agent-based models provide is valuable for modeling and optimization.

## 2 BRANCHING PROCESS ENVIRONMENT

We take a branching process-based view of an infectious disease crisis (Fig. 2). We track two generations of potential individuals: the seed case and their contacts. We assume that interventions begin after a reporting and tracing delay. At that point, day $t_{start}$ ($t_{start} = 3$ in Fig. 2), we observe the symptom history for each agent up to day $t$ and must decide which action to take for each agent (e.g., quarantine, test). On day $t$, we observe the symptom state of each agent plus the results of any sensing actions (defined below) we have taken up to day $t$ and must decide what action to take for each agent on day $t$. The simulation proceeds for a fixed period of time until $T$.

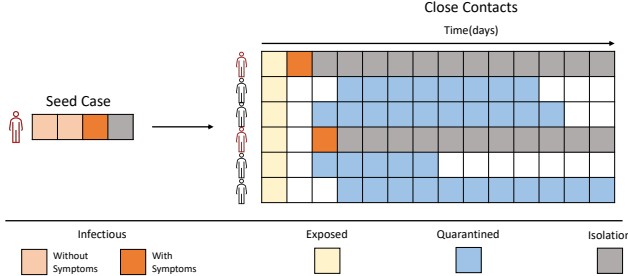

**Figure 2: An agent-based branching process model. The diagram depicts standard contact tracing for an example seed case with six contacts.**

In Fig. 2, we present an application of contact tracing policy in the branching process framework. The seed case remains infectious for two days without exhibiting symptoms, followed by one day with symptoms, before entering isolation. In this example, all six contacts were exposed on the same day. Contacts 1 and 4 are infected and show symptoms on day 2 and day 3, respectively. All contacts are asked for quarantine if their infection probability is higher than a threshold. Contact 3 and contact 5 serve quarantine on day 3. Contact 2 and contact 6 start quarantining on day 4.

In an infectious disease crisis, we can use whatever data is available to construct such a branching process model. Many of the required components are distributions that are often estimated by epidemiologists in the early stages of an outbreak. We describe distributions we used to simulate SARS-CoV-2 and their sources in Tab. 1. Components that are not known can be filled in conservatively or sensitivity analysis can be performed. In some cases, distributional estimates can be shared across diseases—for example, POLYMOD [14] provides contact distributions for the US and Western European settings for both droplet and physical contact. The superspreading dynamics of infection can be impactful because it is often that most transmission is driven by a small number of seed cases, and this concentration can be exploited by control policies [17]. Nevertheless, superspreading dynamics are often poorly understood, especially early in a crisis and greater understanding would benefit approaches such as this paper's.

We define the objective function as

$$(-S_1 - \alpha_2 \times S_2 - \alpha_3 \times S_3)/\textsc{cluster\_size} \tag{1}$$

where

- $S_1$ is the count of *transmission days* where an infected individual is not quarantined,
- $S_2$ is the count of days where a quarantined individual is not infected, and $\alpha_2$ (which we assume is in $[0, 1]$) is the weight for this term,
- $S_3$ is the sum of the action costs (e.g., test cost) and $\alpha_3$ is the weight for this term, and
- $\textsc{cluster\_size}$ normalizes the objectives to a score per individual.

In summary, the objective function seeks to minimize the number of transmission days (i.e., days where an individual is infectious

**Table 1: Parameters of the SARS-CoV-2 branching process model**

| Parameter | Assumed value | Details and references |
|---|---|---|
| Incubation time | Log-normal: Log mean 1.57 days and log std 0.65 days | Mean: 5.94 days. Bi et al. [2] |
| Duration of infectious period | 7 days—2 days before and 5 days after onset if symptomatic | Bi et al. [2] |
| Probability that an infected individual shows symptoms | 0.8 | Buitrago-Garcia et al. [3] |
| Probability of symptoms without infectiousness | 0.01 per day | Perrault et al. [17] |
| Probability of asymptomatic infection | 0.2 | Buitrago-Garcia et al. [3] |
| Probability of highly transmissive | 0.109 | Perrault et al. [17] |
| Infectiousness multiplier for highly transmissive individuals | 24.4 | Perrault et al. [17] |
| Test parameters | TP = 0.86, FP = 0.66 TN = 0.14, FN = 0.34 | Besutti et al. [1] |
| Delays | Observation Delay = 3 days Test Result Delay = 1 day | Assumed |

but not quarantined), minimize the number of days of non-effective quarantine, and minimize the cost associated with actions.

We consider two action types. *Quarantine-type* actions reduce the number of transmission days for an agent. The simplest quarantine-type action causes an agent to not produce a transmission day with probability 1 and incurs no additional cost. A more complex quarantine-type action may work probabilistically (because an individual may not choose to quarantine if directed), incur an additional cost (e.g., the cost of checking in with that individual by phone), or may be coupled with a sensing action (see below). Quarantine-type actions are that they contribute to $S_2$ if the individual quarantines and is not infected.

*Sensing-type* actions do not directly affect the number of transmission days directly. Instead, they reveal information about an individual's infectious state according to a probability distribution. For example, if someone has had known exposure to someone infected, but he/she doesn't show the symptoms. With antigen tests, we can know whether this person is infected or not. Actions can combine both sensing and quarantine, e.g., an action that performs an antigen test and then quarantines if the result is positive.

## 3 APPROACH

We show that the optimization problem from the previous section can be formulated as a partially observable Markov decision process (POMDP). However, solving this POMDP directly is wildly intractable. Some hope arrives from the result that, under a simplified model that contains only sensing-type actions, the POMDP can be solved optimally if the probability that an individual is infectious can be estimated—itself a challenging problem due to the high dimensional observation space.

Motivated by this conclusion, we formulate our solution approach: we use a convolutional neural network (CNN) to estimate the probability of infectiousness for each individual in a cluster, and this output, along with cluster-wide statistics, serves as the state for the RL agent.

### 3.1 POMDP Formulation

We define a POMDP [6] as $\langle S, A, R, P, \Omega, O, \gamma, S_0 \rangle$, where $S$ and $A$ represent the state and action spaces, respectively, $R : S \times A \to \mathbb{R}$ is the reward function, $P : S \times A \to \Delta S$ is the transition function, $\Omega$ is the observation state, $O : S \times A \to \Delta\Omega$ is the observation probabilities, $\gamma \in [0, 1]$ is the discount factor, and $S_0 : \Delta S$ is the distribution of initial states.

We briefly describe how to interpret the control problem of the previous section as a POMDP. We define the state space as containing all of the relevant information required to simulate the cluster, including whether the seed case is highly transmissive, whether each contact of a seed case will become infected, whether they will show symptoms and if so, on what day. This simulator data cannot be observed directly—instead we must rely on receiving action-dependent observations. We define the action space as the set of daily quarantine and sensing actions that are available for each individual in the cluster. For instance, in our experiments, we consider five actions: no quarantine and no test, quarantine and no test, test and no quarantine, test and quarantine, and test and quarantine only if positive. If we have $N$ individuals in the cluster, we have an action space of size $|A|^N$. For observations, we receive two types of information from each individual in each timestep: symptom information and test results. We receive test results only when a sensing-type action is taken and these results are noisy (Tab. 1). Similarly, we always observe symptoms if they are present, but both infectiousness without symptoms and symptoms without infectiousness are possible. The resulting observation space size is $4^N$.

In principle, solving the POMDP formulation results in the optimal control policy. In practice, exact solving is not possible due to the high computational complexity of the best-known algorithms. A particular source of difficulty is the problem of calculating the posterior probability of infection for each individual given the observations. A key challenge is that the variation in infectiousness of the seed case causes the posterior probability of infection for each

individual to depend on the observations for all other individuals. Intuitively, observing symptoms or positive test results for one individual makes it more likely that the seed case is highly transmissive and thus more likely that each other individual is infected.

## 3.2 Optimal Policy Without Sensing Actions

We first consider a simplified POMDP where the only actions available are a quarantine action and no quarantine action. We show that, if the posterior probability of infection can be calculated exactly, the optimal policy has a *threshold-type* form: if the posterior probability of infection is above a threshold, we quarantine and otherwise do not. We show this initially for a costless quarantine action with 100% efficiency as this is what we use in experiments (Thm. 1). We then generalize the result to any menu of non-sensing actions because the expected reward of each action can be exactly calculated given the posterior probability of infection (Thm. 2). We remark that these results provide additional context to the findings of Perrault et al. [17] by defining the class of optimal risk-based policies.

Let $p_{\text{inf}}$ represent the posterior probability of infection for an individual given the observations so far.

THEOREM 1. *With a costless quarantine action that is always successful and a null action, the objective function of Eq. 1, the optimal policy is to quarantine if $p_{inf} > \frac{\alpha_2}{1+\alpha_2}$ and take the null action otherwise.*

PROOF. Because we have access to the exact posterior probability of infection, we can calculate the expected objective value for each action exactly:

$$\mathbb{E}[r] = \begin{cases} -\alpha_2 \cdot (1 - p_{\text{inf}}) & \text{if quarantined} \\ -p_{\text{inf}} & \text{if not quarantined.} \end{cases} \quad (2)$$

We can then show that if $p_{\text{inf}} > \frac{\alpha_2}{1+\alpha_2}$, the quarantine action has higher expected reward. □

We can use the above proof technique to derive the optimal policy for any menu of non-sensing actions. A useful generalization is when the quarantine action has a cost and a failure rate.

THEOREM 2. *With a quarantine action with success rate $0 \le \beta \le 1$ and cost 1 and a null action, the optimal policy is to quarantine if $p_{inf} > \frac{\alpha_2 \cdot \beta + \alpha_3}{(1+\alpha_2) \cdot \beta}$ and otherwise do not.*

These results highlight the importance of the posterior probability of infection. We next dedicate our attention to producing useful estimates of $p_{\text{inf}}$.

## 3.3 Supervised Learning

We could use RL directly to solve the POMDP using the observation information as the state. Indeed, we show that this is somewhat effective if we leverage the state representation we develop in the next section. However, as we know the unobserved infectious state for each agent in simulation, we hypothesize that using a supervised learning model to predict $p_{\text{inf}}$ and using this as input to the RL algorithm will lead to better objective values compared to pure RL (and in the experiments, we see that the improvement is often substantial). Another option for estimating $p_{\text{inf}}$ would be to use an algorithm for approximate probabilistic inference such as Markov

chain Monte Carlo, but doing so is challenging due to the high dimensional discrete observation space where most observations have zero probability for a given state of infectiousness.

A key question for applying supervised learning is how to represent the observation space. We have two desiderata. First, we would like the representation to not vary with cluster size. We can also achieve this property in the RL agent, resulting in an agent that simultaneously be deployed across all cluster sizes, which makes both training and deployment simpler. Second, there is an advantage to using a representation that inherently accounts for the symmetries that arise due to the ordering of individuals, i.e., if we permute the order of individuals in an observation, it should not affect $p_{\text{inf}}$ for each individual.

After testing several representations that satisfy these properties, we arrive at the $7 \times T$ matrix shown in Fig. 3, where $T$ is the simulation length (in our experiments, $T = 30$). This is an egocentric representation of the observation—it is from the perspective of a particular contact and contains all information gathered so far. We train the supervised learning model $f$ to produce output dimension $[0, 1]^T$, i.e., for every day of the simulation, what is the probability that the agent will be infectious given the observation using simulation outputs where the infectiousness of each individual is provided.

The representation contains the following information. The first row is 1 for each day after (inclusive) that the individual shows symptoms. The second row is a binary indicator of whether this day is in the future (1 if yes). The third row is a count of the number of individuals in the cluster that have shown symptoms up to (inclusive) day $t$. The fourth row is the total number of contacts in the cluster minus 1 (constant across time). The fifth row is $t$. The sixth row is 1 if a test was conducted for this individual, and the sixth row represents the results of that test (with a one-day delay). In row 2, 0s are used to indicate that observation was made by this day and 1s represent the future. In row 6 and 7, 0s are used to represent the future (no test was ordered and no results were received).

We will show that this representation can achieve an AUC of 0.95 to predict infectiousness for our branching process model if an appropriate architecture is selected.

| 0 | 1 | 1 | 1 | ... | Symptoms shown by day t? |
|---|---|---|---|-----|--------------------------|
| 0 | 0 | 0 | 1 | ... | 0 for past and present, 1 for future |
| 3 | 3 | 3 | 3 | ... | Total symptom count in cluster |
| 9 | 9 | 9 | 9 | ... | Cluster Size - 1 |
| 0 | 1 | 2 | 3 | ... | t |
| 0 | 1 | 1 | 0 | ... | Test on day t? |
| 0 | 0 | 1 | 0 | ... | Day t-1 test positive? |

**Figure 3: The observation representation used for supervised learning, shown on a cluster of size 10 after observing the outcome of day 2.**

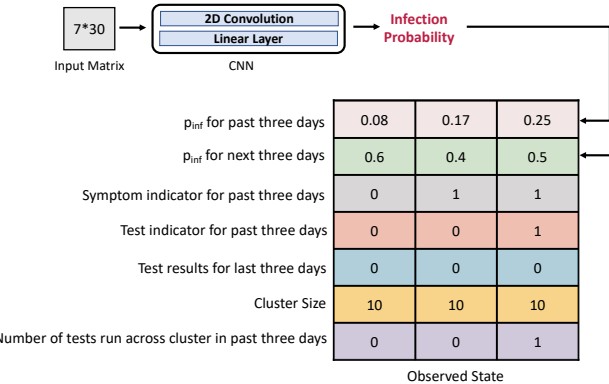

**Figure 4: The supervised learning (CNN) output is used as input to the RL state which prioritizes immediately relevant information.**

## 3.4 Reinforcement Learning

To make RL effective, we develop a compact state representation that includes supervised learning outputs. As with supervised learning, we want the representation to have the same size for all clusters and to naturally encode permutation invariance. The representation we use is a $7 \times 3$ matrix shown in Fig. 4. As with the suprvised learning representation, it is egocentric and time-specific.

The first and second rows represent the $p_{\mathrm{inf}}$ outputs from supervised learning for the last three days and next three days, respectively. The third row indicates whether the individual exhibited symptoms for each day in the past three days. The fourth row is an indicator for if this individual was tested for each of the past three days. The fifth row denotes the test results with a one day delay. The sixth row is the cluster size. The last row indicates the number of tests conducted in the cluster in the past three days.

Training the RL algorithm is straightforward. First, we train the supervised learning predictor from data collected from the simulator. In our experiments, we use a fixed but stochastic control policy to collect this data. This has the advantage that a single supervised learning training run can serve as input to an arbitrary number of RL training. If the optimal policies are dramatically different than the data collection policy, an addition run of supervised learning training can be performed with the current RL policy to increase its accuracy.

Once the supervised learning predictor is trained, we train RL with Proximal Policy Optimization (PPO) [19]. In our experiments, we use six different policy initializations, train each for 800000 environment interactions and pick the best based on 100 evaluation runs. All training is performed on a single core, using Intel i5-8259U @2.3GHz with 8GB of RAM, and a single RL training run takes 20 minutes.

## 4 EXPERIMENTS

We compare different control policies in the branching process environment we construct for SARS-CoV-2. We consider a set of five control actions for each individual for each day: null action, quarantine, test but don't quarantine, quarantine but don't test, and

test and quarantine only if results are positive. We assume that there is no failure rate for actions, and all actions that include a test cost 1 and others are costless. For $\alpha_2$, we use small values of 0.01 and 0.02 as typical SARS-CoV-2 contact tracing policies accept a large number of quarantine days for non-infectious individuals. For $\alpha_3$, we use values of 0.001, 0.005, 0.01, 0.02, 0.03 and 0.2. We sample cluster size from a uniform distribution on (2, 40). The model code is available online (https://github.com/XueqiaoPeng/CovidRL).

### 4.1 Supervised Learning Model

We experiment with a variety of supervised learning model architectures (Tab. 2) to find one that achieves a high AUC across cluster sizes. We find that CNNs are generally most effective and compare different kernels and layer structures. In single layer architectures, we find that larger 2D convolutions tend to achieve higher AUC. We then found that a single convolution layer followed by a linear layer performs just as well as deeper architectures—this setup of a (5, 2) 2D convolution followed by a linear layer is what we use in the experiments below.

**Table 2: We find that two-layer architectures using a 2D convolution followed by a linear layer achieve performance on par with larger models.**

|         |                                                  | Cluster size = 4 | 8     | 16    | 32    |
|---------|--------------------------------------------------|------------------|-------|-------|-------|
| 1 Layer | Conv1d (5,2)                                     | 0.798            | 0.807 | 0.823 | 0.830 |
|         | Conv1d (5,3)                                     | 0.814            | 0.830 | 0.835 | 0.839 |
|         | Conv2d (5,2)                                     | 0.800            | 0.814 | 0.827 | 0.830 |
|         | Conv2d (5,3)                                     | 0.832            | 0.820 | 0.838 | 0.840 |
|         | Conv2d (5,4)                                     | 0.858            | 0.849 | 0.843 | 0.859 |
|         | Conv2d (5,5)                                     | 0.864            | 0.895 | 0.893 | 0.893 |
| 2 Layer | Conv1d (5,2) Conv1d (1,2)                        | 0.824            | 0.830 | 0.833 | 0.840 |
|         | Conv2d (5,3) Conv2d (1,3)                        | 0.883            | 0.903 | 0.898 | 0.897 |
|         | Conv2d( 5,2) Linear Layer                        | 0.955            | 0.960 | 0.947 | 0.961 |
|         | Conv2d (5,3) Linear Layer                        | 0.951            | 0.960 | 0.940 | 0.964 |
| 3 Layer | Conv1d (5,3) Conv1d (1,3) Linear Layer           | 0.958            | 0.957 | 0.950 | 0.961 |
| 4 Layer | Conv1d (4,3) Conv1d (2,3) Conv1d (1,3) Linear Layer | 0.958         | 0.958 | 0.953 | 0.965 |

### 4.2 Benchmark Policies

We compare the RLSL approach we propose to several baselines.

- **Threshold** is the threshold-type policy suggested in Sec. 3.2. It does not use test actions. This policy turns out to be highly conservative and results in long quarantine duration for all contacts for the tested $\alpha_2$ values.

**Table 3: RLSL achieves higher objective values (higher is better) than baselines across all tested $\alpha_2$ and $\alpha_3$.**

| | $\alpha_2 = 0.01$ $\alpha_3 = 0.001$ | $\alpha_2 = 0.01$ $\alpha_3 = 0.005$ | $\alpha_2 = 0.01$ $\alpha_3 = 0.01$ | $\alpha_2 = 0.01$ $\alpha_3 = 0.02$ | $\alpha_2 = 0.01$ $\alpha_3 = 0.03$ | $\alpha_2 = 0.01$ $\alpha_3 = 0.2$ | $\alpha_2 = 0.02$ $\alpha_3 = 0.001$ | $\alpha_2 = 0.02$ $\alpha_3 = 0.005$ | $\alpha_2 = 0.02$ $\alpha_3 = 0.01$ | $\alpha_2 = 0.02$ $\alpha_3 = 0.02$ | $\alpha_2 = 0.02$ $\alpha_3 = 0.03$ | $\alpha_2 = 0.02$ $\alpha_3 = 0.2$ |
|---|---|---|---|---|---|---|---|---|---|---|---|---|
| **RLSL (Ours)** | **−3.77 ± 0.25** | **−10.27 ± 0.15** | **−17.13 ± 0.48** | **−44.22 ± 0.84** | **−46.46 ± 1.47** | **−110.92 ± 1.54** | **−4.01 ± 0.21** | **−17.64 ± 0.32** | **−25.39 ± 0.48** | **−49.28 ± 0.66** | **−64.45 ± 0.83** | **−120.21 ± 0.22** |
| Threshold | −21.79 ± 0.20 | −21.79 ± 0.20 | −21.79 ± 0.20 | −21.79 ± 0.20 | −21.79 ± 0.20 | −21.79 ± 0.20 | −43.65 ± 0.32 | −43.65 ± 0.32 | −43.65 ± 0.32 | −43.65 ± 0.32 | −43.65 ± 0.32 | −43.65 ± 0.32 |
| Symptom-Based Quarantine | −111.13 ± 14.18 | −111.13 ± 14.18 | −111.13 ± 14.18 | −111.13 ± 14.18 | −111.13 ± 14.18 | −111.13 ± 14.18 | −112.60 ± 11.94 | −112.60 ± 11.94 | −112.60 ± 11.94 | −112.60 ± 11.94 | −112.60 ± 11.94 | −112.60 ± 11.94 |
| 14 Days Quarantine | −97.18 ± 9.97 | −97.18 ± 9.97 | −97.18 ± 9.97 | −97.18 ± 9.97 | −97.18 ± 9.97 | −97.18 ± 9.97 | −106.63 ± 11.00 | −106.63 ± 11.00 | −106.63 ± 11.00 | −106.63 ± 11.00 | −106.63 ± 11.00 | −106.63 ± 11.00 |
| No Quarantine | −235.98 ± 18.53 | −235.98 ± 18.53 | −235.98 ± 18.53 | −235.98 ± 18.53 | −235.98 ± 18.53 | −235.98 ± 18.53 | −242.16 ± 20.38 | −242.16 ± 20.38 | −242.16 ± 20.38 | −242.16 ± 20.38 | −242.16 ± 20.38 | −242.16 ± 20.38 |

**Table 4: $S_1$, $S_2$ and $S_3$ per individual compared across different cluster sizes (lower is better), using $\alpha_2 = 0.01$ and $\alpha_3 = 0.01$. Even relatively conservative strategies such as 14-day quarantine from exposure fail to isolate some infections in our simulation. RLSL can benefit substantially from the additional information available in large clusters resulting in strong performance with low test costs.**

| | Cluster size = 4 | | | Cluster size = 8 | | | Cluster size = 16 | | | Cluster size = 32 | | |
|---|---|---|---|---|---|---|---|---|---|---|---|---|
| | $S_1$ | $S_2$ | $S_3$ | $S_1$ | $S_2$ | $S_3$ | $S_1$ | $S_2$ | $S_3$ | $S_1$ | $S_2$ | $S_3$ |
| **RLSL** | 0.064 ± 0.008 | 6.808 ± 0.184 | 10.144 ± 0.052 | 0.077 ± 0.012 | 7.552 ± 0.099 | 11.825 ± 0.056 | 0.075 ± 0.011 | 10.033 ± 0.127 | 11.253 ± 0.087 | 0.054 ± 0.007 | 8.259 ± 0.090 | 10.808 ± 0.134 |
| Threshold | 0.078 ± 0.013 | 16.012 ± 0.211 | - | 0.063 ± 0.013 | 17.656 ± 0.198 | - | 0.05 ± 0.008 | 19.681 ± 0.173 | - | 0.016 ± 0.003 | 20.701 ± 0.319 | - |
| Symptom-Based Quarantine | 1.418 ± 0.199 | 0.236 ± 0.029 | - | 1.207 ± 0.187 | 0.239 ± 0.014 | - | 1.196 ± 0.052 | 0.232 ± 0.017 | - | 1.072 ± 0.146 | 0.261 ± 0.042 | - |
| 14-day Quarantine | 1.042 ± 0.072 | 2.469 ± 0.113 | - | 0.965 ± 0.082 | 2.440 ± 0.144 | - | 0.973 ± 0.114 | 2.291 ± 0.125 | - | 0.929 ± 0.107 | 2.004 ± 0.155 | - |
| No Quarantine | 2.361 ± 0.195 | - | - | 2.597 ± 0.282 | - | - | 2.075 ± 0.203 | - | - | 1.856 ± 0.173 | - | - |

**Table 5: In cases where test costs are higher, RLSL produces polices that test too often, resulting in lower performance than RLSL models with only quarantine actions—we discuss potential fixes.**

| | $\alpha_2 = 0.01$ $\alpha_3 = 0.001$ | $\alpha_2 = 0.01$ $\alpha_3 = 0.005$ | $\alpha_2 = 0.01$ $\alpha_3 = 0.01$ | $\alpha_2 = 0.01$ $\alpha_3 = 0.02$ | $\alpha_2 = 0.01$ $\alpha_3 = 0.03$ | $\alpha_2 = 0.01$ $\alpha_3 = 0.2$ | $\alpha_2 = 0.02$ $\alpha_3 = 0.001$ | $\alpha_2 = 0.02$ $\alpha_3 = 0.005$ | $\alpha_2 = 0.02$ $\alpha_3 = 0.01$ | $\alpha_2 = 0.02$ $\alpha_3 = 0.02$ | $\alpha_2 = 0.02$ $\alpha_3 = 0.03$ | $\alpha_2 = 0.02$ $\alpha_3 = 0.2$ |
|---|---|---|---|---|---|---|---|---|---|---|---|---|
| RLSL | **−3.77 ± 0.25** | **−10.27 ± 0.15** | **−17.13 ± 0.48** | **−44.22 ± 0.84** | **−46.46 ± 1.47** | **−110.92 ± 1.54** | **−4.01 ± 0.21** | **−17.64 ± 0.32** | **−25.39 ± 0.48** | **−49.28 ± 0.66** | **−64.45 ± 0.83** | **−120.21 ± 0.22** |
| RLSL (Daily Test) | −4.30 ± 0.42 | −13.15 ± 0.15 | −24.46 ± 0.17 | −45.62 ± 1.27 | −74.68 ± 0.2 | −737.78 ± 3.33 | −12.81 ± 0.55 | −23.72 ± 0.47 | −27.25 ± 0.58 | −50.50 ± 0.11 | −75.88 ± 0.26 | −739.98 ± 1.516 |
| RLSL (No Test) | −34.56 ± 0.39 | −34.56 ± 0.39 | −34.56 ± 0.39 | −34.56 ± 0.39 | −34.56 ± 0.39 | −34.56 ± .39 | −52.92 ± 0.13 | −52.92 ± 0.13 | −52.92 ± 0.13 | −52.92 ± 0.13 | −52.92 ± 0.13 | −52.92 ± 0.13 |
| RL Only | −14.64 ± 0.79 | −20.32 ± 0.83 | −34.02 ± 0.70 | −46.10 ± 1.14 | −53.22 ± 1.01 | −84.35 ± 1.04 | −15.36 ± 0.76 | −25.66 ± 0.56 | −39.80 ± 0.39 | −63.07 ± 0.81 | −70.56 ± 0.827 | −162.4 ± 2.36 |
| Threshold (SL Only) | −21.79 ± 0.20 | −21.79 ± 0.20 | −21.79 ± 0.20 | **−21.79 ± 0.20** | **−21.79 ± 0.20** | **−21.79 ± 0.20** | −43.65 ± 0.32 | −43.65 ± 0.32 | −43.65 ± 0.32 | **−43.65 ± 0.32** | **−43.65 ± 0.32** | **−43.65 ± 0.32** |

- **Symptom-Based Quarantine** quarantines if an individual exhibits symptoms on the day before the observed day and otherwise does not.
- **14-Day Quarantine** quarantines individuals from the initial day they exhibit symptoms until either 14 days have passed or until they no longer exhibit symptoms, whichever is later. No test action is included.
- **No Quarantine** always performs the null action.

## 4.3 Analysis

Our experimental results report the average objective value and standard error taken over 10 random clusters (Tab. 3). We find that RLSL and Threshold acheive better performance than baselines in all cases. However, our current methods for RLSL struggle relative to Threshold when tests are expensive. Our experimental results could be broadened by including more $\alpha$ values and more analysis as to where the RLSL policies gain their advantage (but see discussion of Tab. 5 below for some insights).

Focusing on the setting of $\alpha_1 = 0.01$ and $\alpha_2 = 0.01$, we report objective values broken out by component and by cluster size as measured per individual (Tab. 4). Here we can get an intuitive grasp of what is happening in the different policies. Threshold aggressively quarantines, resulting in $S_2 = 16$–$20$, i.e., 16–20 days of quarantine without infection per contact, for the tested $\alpha$ values. This is able to drive $S_1$ to a low value, resulting in an average objective value of −21.79. Recall that $S_1$ is much more highly weighted

(100 times) higher than $S_2$ in this setting. Symptom-based and 14-day quarantine reduce $S_2$ by a factor of 8 to 100, but this causes $S_1$ to be roughly 150 to 200 times higher. By leveraging tests, RLSL can reduce $S_2$ by a factor of 2–3 and $S_1$ by a factor of 0.8–3.5.

In the ablation study (Tab. 5), we gain a more detailed view into the operation of the RLSL policy. We see that the introduction of the SL outputs to the RL state results in better performance in all tested scenarios compared to RL Only, which uses the state representation of Fig. 4 without the first two rows.

We can observe limitations of the supervised infectiousness prediction model in Tab. 4, where the $S_2$ cost does not decrease as cluster size increases—from Thm. 1, we can conclude that if $p_{\text{inf}}$ is correct, the ratio of $S_1$ to $S_2$ should not depend on cluster size for Threshold. There are several possible causes of this issue. First, the SL model outputs might be miscalibrated, as is often the case for neural networks trained on highly imbalanced data. This issue could be fixed with post-hoc calibration such as Platt scaling [18]. In this instance, a more sophisticated calibration could be employed with separate calibration parameters per cluster size, if necessary. Second, it may be the case that the SL model outputs are wrong for reasons other than calibration. For example, it may receive insufficient relevant training data as it is trained on data produced from a random policy and not Threshold or RLSL. It is also possible that we performed insufficient architecture search.

We also see that RLSL (No Test) often performs better than RLSL as test costs increase. This suggests that RLSL is not finding a true

optimal policy. This could likely be address by using a wider range of initialization values for RLSL—for example, initializing some seeds to policies that test very little (the initialization we use for RLSL and RL Only tests heavily). This observation has a silver lining: RL (No Test) can achieve much stronger performance than baselines even without tests. This implies that RL (No Test) is able to correct for the errors in Threshold to find a policy closer to what is suggested by Thm. 1.

## 5 DISCUSSION AND FUTURE WORK

This work aims to develop a generic multi-objective optimization approach for cluster-level optimization of NPIs. We formulate this problem for RL in a branching process environment. We present initial results that demonstrate the potential of our approach—in a branching process model of SARS-CoV-2, we can achieve substantially higher objective values than baseline policies. The resulting policies can be applied across all cluster sizes and do not take much time to train on consumer hardware. The policies we propose are able to heavily exploit superspreading dynamics.

Our vision for an infectious disease crisis is that a canonical probabilistic model of the disease is constructed and updated throughout the crisis. The model can be constructed from estimates of key disease parameters that are made from various sources throughout a crisis and can reflect uncertainty in these estimates. We advocate that superspreading dynamics be given substantial attention in these early stages due to the substantial influence on interventions that we find it can have. Using this canonical model, a branching process environment can be constructed and optimized against as we propose in this paper. We do not consider uncertainty in the parameters of this model, but it is possible to do so with existing techniques and leads to different RL algorithmic choices depending on the form of the uncertainty and the desired objective.

A key disadvantage of our approach as presented is the complexity of the resulting policies. For instance, to execute our RLSL policy requires training and drawing outputs from two neural networks. In contrast, policies that were employed in the SARS-CoV-2 pandemic consisted of short lists of rules. We believe that this is not an inherent weakness of our approach—we can leverage interpretable ML and RL techniques to "distill" the RLSL policies into, say, low-depth decision trees, allowing them to be applied at scale with low logistical cost. There will be some decrease in quality, but we suspect still substantial advantage over baselines.

An area for future study is cost and benefit of taking a cluster- rather than individual-level view of policy application. This imposes additional logistical costs and the benefit is dependent on the degree of cluster-level transmission heterogeneity that is present. This trade-off is not well understood and is a critical area for future work.

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
