# OpenReview forum: "Using Reinforcement Learning for Multi-Objective Cluster-Level NPI Optimization"
_KDD.org/2023/Workshop/epiDAMIK — KDD 2023 Workshop epiDAMIK_

### Official Review · Reviewer_1gDX · 2023-06-27
**Good paper with compelling results, could benefit from more intuition for methodological choices**

**Rating:** 4
**Confidence:** 4

**Review:**

This paper describes a novel RL approach to optimizing infection disease control policy. The proposed method combines supervised learning with RL and shows strong performance compared to baseline policies.

Positives:
+ The branching-process formulation is well-described and intuitive
+ The need for an estimate of the probability of infection is well-motivated
+ The experiments are extensive and clearly show the strengths (and limitations) of the proposed RLSL framework
+ Overall, the paper is clear and easy to follow


Places for Improvements
- The intuition of the representation and state space for both the SL and RL settings could be improved. Currently, it seems as though many representations were tested and this one was eventually chosen. Were they tested on validation data? More information about how these representations are necessary
- Why was PPO chosen for the RL policy? Were other RL techniques considered? This choice could benefit from more justification
- How hyperparameters were chosen should be discussed more. Currently, it almost seems as though the architecture and hyperparameters for the 2D CNN were chosen based on the same set in which policies were evaluated, which would be problematic
- As mentioned in the paper, calibration of the SL estimates is critical for the threshold-based approach. The authors should consider calibrating these probabilities and evaluating the calibration error in some way, to see if it can improve all methods, especially the threshold-based baseline

---

### Official Review · Reviewer_mSgB · 2023-06-29

**Rating:** 3
**Confidence:** 4

**Review:**

### Summary

This work proposes to use reinforcement learning for optimizing multi-objective infectious disease control policy in a branching process environment. In the approach, this paper uses a convolutional neural network to estimate the probability of infectiousness for each individual in a cluster and use the outputs as the state of the RL agent. This work evaluates the proposed approach in a branching process simulated for SARS-CoV-2 and compares the approach with baseline policies. The baselines include thresholding, Symptom-Based Quarantine, 14-Day Quarantine, and no quarantine. The results show that the proposed approach achieves higher objective values than the baselines across multiple parameter settings.

### Weaknesses

- The environment setup needs to be further explained. For example, it would be better to provide formal definitions of the branching process environment, including the states and necessary parameters. Moreover, the example illustrated in Figure 2 is confusing. For example, it would be better to explain what factors cause the state changes in different clusters.
- Further discussion and comparison with related work need to be incorporated. It would be better to provide a more detailed discussion with related work, especially previous decision-making methods or RL methods for optimizing intervention policy. For example, the related work of RLGN [1]. It would be better to compare such methods in the experiments. Moreover, the motivation for using branching processes and cluster-based view needs to be further elaborated.
- More details in the experiments need to be included. For example, the detailed setup of the branching processes for SARS-CoV-2 and its hyper-parameter settings and the details of how training examples are generated. Including such details help better interpret the results of the comparison.

[1] Eli Meirom, Haggai Maron, Shie Mannor, and Gal Chechik. 2021. Controlling graph dynamics with reinforcement learning and graph neural networks. In International Conference on Machine Learning. PMLR, 7565–7577

---

### Official Review · Reviewer_z183 · 2023-06-30
**Interesting problem.. parts of approach unclear**

**Rating:** 2
**Confidence:** 4

**Review:**

The paper aims to learn contact tracing policies by bridging SL and RL. In implementation, a CNN is used to estimate the probability of infectiousness for each individual in a cluster, and this output, along with cluster-wide statistics, serves as the state for the RL agent which learns a cluster-level lockdown policy.
Briefly, first data is sampled from the simulator under arbitrary control policies to predict infectiousness of agents. Once predictor is trained, how the outputs are used to define state of the RL agent to learn cluster-level policies. The cluster sizes are from 8 to 32. The policy is learned over a space of 5 discrete actions over an objective to minimize (number of transmission days + non-effective quarantine + costs).

The problem of interest is very impact and the idea of using RL to learn NPI policies is exciting. However, the current formulation and assumptions seems a bit unrealistic and results are also not very encouraging. I would suggest authors to revisit the experiment design and then resubmit a manuscript.

Some comments and questions to think about:
1. The SL training setup seems highly unrealistic since it uses ground truth not available in the real world. How can the exact infection probability of an individual estimated for ground truth? How will this approach generalize? Even results in Table 5 mean that " SL outputs could be mis-calibrated". Also, it is less intuitive to learn input state parameters for an RL agent when data can be approximated from the the environment (since was used for SL ground-truth)?
2. Effect of decisions on cluster sizes will depend on their relative scale w.r.t the size of the total population. if clusters are as small as 8, 16 or 32 people -- it will be very tough to observe distinction between individual people and clusters. To make claims for real policy decision making: clusters should at-least be a census block [or county] and the simulation should analyze how these variables change with scale and mobility across clusters. What is the size of the total population considered? This was not evident from experiments.

I think these are sensitive problems with far-reaching implications. More research needs to be done before claims are put out into the world.